# BID-LoRA: A Parameter-Efficient Framework for Continual Learning and Unlearning

## Abstract

Recent advances in deep learning underscore the need for systems that can not only acquire new knowledge through Continual Learning (CL) but also remove outdated, sensitive, or private information through Machine Unlearning (MU). However, while CL methods are well-developed, MU techniques remain in early stages, creating a critical gap for unified frameworks that depend on both capabilities. We find that naively combining existing CL and MU approaches results in *knowledge leakage* a gradual degradation of foundational knowledge across repeated adaptation cycles. To address this, we formalize Continual Learning Unlearning (CLU) as a unified paradigm with three key goals: (i) precise deletion of unwanted knowledge, (ii) efficient integration of new knowledge while preserving prior information, and (iii) minimizing knowledge leakage across cycles. We propose Bi-Directional Low-Rank Adaptation (BID-LoRA), a novel framework featuring three dedicated adapter pathways-retain, new, and unlearn-applied to attention layers, combined with escape unlearning that pushes forget-class embeddings to positions maximally distant from retained knowledge, updating only $\approx 5\%$ of parameters. Experiments on CIFAR-100 show that BID-LoRA outperforms CLU baselines across multiple adaptation cycles. We further evaluate on CASIA-Face100, a curated face recognition subset, demonstrating practical applicability to real-world identity management systems where new users must be enrolled and withdrawn users removed.

## 1 Introduction

Recent developments in AI have triggered a new shift in deep learning models Kaplan et al. (2020). Future intelligent systems are expected to follow a dual paradigm: learn new knowledge, and remove specific knowledge. This capability is called Continual Learning and Unlearning (CLU) Chatterjee et al. (2024); Liu et al. (2022), also termed continual adaptation. This capability becomes essential when data distributions shift over time, whether due to policy changes, task modifications, or market trends. Regardless of the underlying cause, the model's knowledge base must be updated accordingly. Fig 1 illustrates CLU system as new data replaces unwanted data. This requires simultaneous systems for both learning and unlearning.

Since CLU combines Continual Learning (CL) and Machine Unlearning (MU), we establish each component's foundation. CL is the process of acquiring new knowledge Kirkpatrick et al. (2017); Buzzega et al. (2020); Caccia et al. (2021); Douillard et al. (2022); Cotogni et al. (2025); Mohamed et al. (2023) from unseen data while retaining performance on existing tasks, widely studied in machine learning. Application tasks include mixed-task scenarios, out of distribution tasks, among others. While CL focuses on knowledge acquisition, MU addresses selective knowledge removal, an emerging field Kurmanji et al. (2023); Zhao et al. (2024); Cha et al. (2024); Fan et al. (2023); Tarun et al. (2023). Targets for unwanted knowledge removal include privacy-related data, regulatory removal by government orders such as General Data Protection Regulation (GDPR) and California Consumer Privacy Act (CCPA) Goldman (2020); Data (2018), and hate speech, among others.

There is an intrinsic connection between MU and CL in which one adds knowledge and other removes knowledge, both retaining existing knowledge. Can we combine both for simultaneous learning and unlearning?

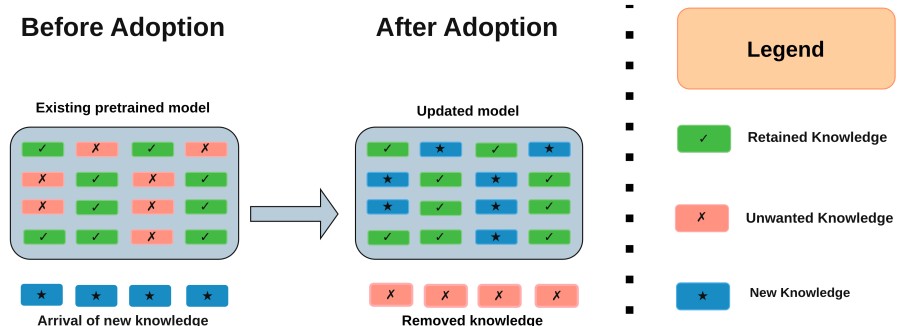

Figure 1: Overview of CLU. The CLU system removes unwanted knowledge (red), retains prior knowledge (green), and integrates new knowledge (blue).

Such a framework would enable access control, surveillance, personalized education, and content moderation. Can continual learning alone solve this? No-without removing unwanted knowledge over time, it accumulates and causes knowledge drift degrading model performance and creating bias toward unwanted information, making unlearning essential.

Despite CLU's importance, a research gap exists due to maturity imbalance. CL is well-established, while Machine Unlearning remains developmental. This disparity leaves unified CLU frameworks largely unexplored. CLU applications demand parameter-efficient solutions. CLU systems continuously adapt as data arrives and require removal. Full retraining becomes computationally infeasible, making parameter-efficient approaches essential. However, combining CL and MU introduces knowledge leakage-repeated learning-unlearning cycles cause models to gradually lose foundational knowledge. Unlike catastrophic forgetting, knowledge leakage is a slow degradation of original capabilities across adaptation cycles; further analysis is provided in Section 6.2. This challenge is critical because real-world systems face continuous requests: employees join and leave organizations, regulations evolve, and user preferences shift constantly. Such systems with minimal knowledge leakage would enable critical applications: language models, medical systems, recommendation engines, and autonomous vehicles. Face recognition presents the most urgent CLU need due to privacy challenges.

Face recognition systems are critical CLU applications as personnel frequently join and leave organizations. Since facial data is highly sensitive private information, protecting it from misuse through model inversion techniques Fredrikson et al. (2015); Zhu et al. (2019); Zhang et al. (2020) becomes essential to safeguard privacy rights mandated by GDPR and CCPA Goldman (2020); Data (2018). Face recognition thus provides a natural testbed for CLU applications, presenting an urgent challenge that demands immediate attention given the escalating privacy threats and regulatory requirements. Designing such an ideal CLU system presents three core challenges. First, achieving continual learning while avoiding catastrophic forgetting. Second, implementing selective machine unlearning. Finally, maintaining knowledge stability and minimizing knowledge leakage across repeated cycles.

To address these challenges, we propose Bi-Directional LoRA (BID-LoRA), a parameter-efficient solution to CLU problems that minimizes knowledge leakage. BID-LoRA leverages LoRA's inherent parameter efficiency, fine-tuning only attention layers in transformer blocks and classification heads Houlsby et al. (2019); Hu et al. (2022); Li & Liang (2021); Geva et al. (2020) as fine-tuning fewer parameters has been shown effective Mallya & Lazebnik (2018); Mallya et al. (2018); Wang et al. (2022); Zhang et al. (2022) in knowledge manipulation. To reduce catastrophic forgetting Kirkpatrick et al. (2017) and minimize knowledge leakage, we employ the replay mechanism. This approach is similar to performing minimally invasive surgery on a model rather than major surgery. BID-LoRA is simple, parameter-efficient, data-efficient, and suitable for large models. We conduct extensive experiments across classification and face recognition tasks, demonstrating broad applicability. Our contributions are summarized as follows:

- We formally define the Continual Learning-Unlearning (CLU) problem and demonstrate that naively combining existing CL and MU approaches leads to significant knowledge leakage, where foundational knowledge degrades across repeated adaptation cycles.

- We propose BID-LoRA, a parameter-efficient framework featuring three-pathway separation that isolates retained, forgotten, and newly learned knowledge. This architecture prevents interference between competing objectives while updating only $\approx 5\%$ of model parameters.

- We introduce escape unlearning, which computes optimal embedding locations that are maximally distant from retained class centroids, pushing forget-class representations to positions that are both hard to recover and non-interfering with retained knowledge.

- We establish the first generalizable CLU benchmark through comprehensive experiments on CIFAR-100 and CASIA-Face100, employing a novel sliding window evaluation protocol with progressive 10-class forget-retain-learn cycles that systematically tests long-term adaptation stability.

- We demonstrate BID-LoRA's practical applicability to face recognition and identity management systems, where privacy-driven unlearning (e.g., GDPR compliance) must coexist with incremental enrollment of new users.

## 2 Related work

### 2.1 Continual Learning

Continual Learning (CL) aims to train models sequentially on evolving tasks while retaining prior knowledge. In the CL setting, new classes arrive without access to old data, often leading to catastrophic forgetting. To address this, several strategies have emerged.

Kirkpatrick *et al.* Kirkpatrick et al. (2017) introduced Elastic Weight Consolidation (EWC), an early exemplar-free approach that adds a Fisher-based quadratic penalty to protect parameters critical for past tasks. Buzzega *et al.* Buzzega et al. (2020) proposed Dark Experience Replay++ (DER++), combining rehearsal with distillation by storing both samples and logits to stabilize decision boundaries. Caccia *et al.* Caccia et al. (2021) developed ER-ACE and ER-AML where cross-entropy is applied asymmetrically, isolating new-class updates while replay consolidates all classes. Douillard *et al.* Douillard et al. (2022) presented DyTox, a transformer-based method using task-specific tokens and a replay buffer to scale across tasks without task IDs. Cotogni *et al.* Cotogni et al. (2025) introduced GCAB, which employs gated class attention and feature drift compensation for exemplar-free ViT learning. Finally, Mohamed *et al.* Mohamed et al. (2023) proposed D3Former, which debiases logits and preserves attention maps to balance performance across old and new classes.

### 2.2 Machine Unlearning

Machine Unlearning (MU) has seen significant advancements, focusing on methods that effectively balance the removal of data influence and the preservation of retained knowledge. Kurmanji *et al.* Kurmanji et al. (2023) tackled the problem of unlearning by pushing the forget distribution toward a uniform distribution, while ensuring that the retain distribution follows the normal loss function. This approach ensures that models forget the target data without significantly compromising performance on the remaining data. Zhao *et al.* Zhao et al. (2024) addressed the issue of continual forgetting in the context of continual learning by utilizing GS-LoRA, which targets the FFN modules with a group sparsity regularizer. Cha *et al.* Cha et al. (2024) introduced adversarial examples as a technique to ensure a cleaner and more robust removal of forget data. By generating adversarial examples, the model is encouraged to misclassify the forget data, making it harder for the model to remember or retain the forgotten samples. Fan *et al.* Fan et al. (2023) proposed a novel method that computes the weight saliency matrix for data to be forgotten. By identifying the most relevant weights for the forgotten samples, the model updates only those weights, which helps to avoid catastrophic forgetting and ensures that the model retains critical information from the retained data. Tarun *et al.* Tarun et al. (2023) proposed a method that corrupts the knowledge to be forgotten by using

noise and then performs repair work to maintain the retained knowledge. Panda *et al.* Panda & Prathosh (2024) proposed a weak unlearning approach for black-box GANs that filters undesired outputs by computing projection similarity between sampled latent vectors and a learned representation of unwanted features in the latent space. Wang *et al.* Wang et al. (2025) proposed MCC-Fed, which detects malicious clients via Euclidean distance-based deviation analysis and employs a Lipschitz-inspired contribution-aware metric as a regularization term to precisely unlearn their negative influence without requiring auxiliary datasets.

### 2.3 Parameter-Efficient Fine-Tuning

Fine-tuning large pretrained models such as vision and language models on downstream tasks has become a dominant paradigm in modern deep learning. Parameter-efficient fine-tuning techniques are widely adopted, as they substantially reduce trainable parameters without significant performance degradation. Addition-based approaches Liu et al. (2024), which introduce new trainable components while freezing the original model, other works include Houlsby et al. (2019); Li & Liang (2021). Freezing-based techniques Lee et al. (2019) represent another approach, selectively updating only specific parameters or layers while keeping the rest frozen. Parameter factorization methods Chavan et al. (2023); Valipour et al. (2022) form the third category, decomposing weight updates into low-rank matrices to achieve efficiency. Among these, Low-Rank Adaptation (LoRA) Hu et al. (2022) has emerged as particularly effective, decomposing weight updates into low-rank matrices that can be efficiently trained and merged. Our work builds upon LoRA's foundation to address the unique challenges of continual learning-unlearning.

### 2.4 Continual Learning-Unlearning (CLU)

The existing CLU works, such as Shibata *et al.* Shibata et al. (2021), Liu *et al.* Liu et al. (2022), Chatterjee *et al.* Chatterjee et al. (2024), and Huang *et al.* Huang et al. (2025), integrate CL and MU, providing theoretical foundations for adaptive knowledge management. While this field has gained attention, existing approaches still face significant design and efficiency challenges.

Shibata *et al.* Shibata et al. (2021) first explored a CLU-like system using mnemonic codes, enabling selective forgetting by discarding class-specific codes and new learning by embedding fresh ones. However, this method requires retraining from scratch with mnemonic embeddings, is incompatible with pre-trained models, and doubles computational cost. Liu *et al.* Liu et al. (2022) proposed CLPU-DER++, which achieves exact unlearning via isolated temporary networks that can be deleted while retaining a permanent model. It guarantees privacy and supports selective forgetting without original data. Yet, it demands retraining from scratch, incurs exponential storage overhead with multiple tasks, and cannot revise permanent knowledge. Chatterjee *et al.* Chatterjee et al. (2024) introduced UniCLUN, a dual-teacher distillation framework with one CL teacher, one UL teacher, and a student model. It is the first unified CL–UL approach, handling mixed learning–forgetting sequences adaptively. Nevertheless, it incurs parameter overhead, requires heavy computation, and depends on replay buffers that raise privacy risks. Adhikari *et al.* Adhikari et al. (2025) proposed UnCLe, a hypernetwork-based framework that generates task-specific parameters conditioned on task embeddings, enabling data-free unlearning by aligning forget-task parameters with Gaussian noise. However, it supports only task-level unlearning and cannot be directly applied to pretrained models like DeiT, as it requires training a hypernetwork to generate entire model parameters from scratch rather than fine-tuning existing weights. More recently, Huang *et al.* Huang et al. (2025) proposed a gradient-based, task-agnostic CLU framework optimized via KL divergence, aiming for efficiency and adaptability. However, it requires computing online Hessian approximations through costly inner-loop optimization, lacks support for pretrained vision transformers, and has only been validated on models trained from scratch rather than leveraging existing foundation models.

Despite these contributions, none of the above methods are resource-friendly: most update nearly all parameters, making them inefficient for large-scale pre-trained models. Moreover, they have only been tested on limited scenarios involving a few additions and deletions, raising questions about their claims of 'continual' operation. In contrast, our BID-LoRA is explicitly designed to be parameter-efficient, resource-conscious, and validated across long sequences of CLU requests, demonstrating genuine continual adaptability at scale.

# 3 Problem Formulation

## 3.1 Problem Setting

We introduce a novel problem setting called Continual Learning Unlearning (CLU) also known as Continual Adaptation or simply Adaptations, a framework for dynamically modifying model knowledge defined as learned class mappings encoded in parameters. This setting brings together two complementary goals: (i) selectively removing specific targeted knowledge (machine unlearning) and (ii) acquiring new targeted knowledge for existing pre-trained models (continual learning), all while maintaining performance on the remaining knowledge in a continual form. To develop a solution for this problem, we first present the most basic case, in which only one adaptation task is performed, and then expand this formulation to the continuous scenario, which includes a series of adaptation activities.

Let $\mathcal{M}$ be a model pre-trained on the dataset $D$. We regard $\mathcal{M}$ as a mapping function $f_{\mathcal{M}} : \mathcal{X}_D \rightarrow \mathcal{Y}_D$, where $\mathcal{X}_D$ and $\mathcal{Y}_D$ denote the input and output spaces associated with $D$, respectively. Our objective is to selectively discard certain knowledge while adding new knowledge and retaining the rest. We assume $D_f$ be the dataset to be forgotten, $D_r^{full}$ the dataset to be retained, and $D_{\text{new}}$ the new dataset to be learned, satisfying $D = D_r^{full} \cup D_f$, $D_r^{full} \cap D_f = \emptyset$, and $D_{\text{new}} \cap D = \emptyset$.

In practice, storing the full retain set is prohibited under General Data Protection Regulation (GDPR) and California Consumer Privacy Act (CCPA) regulations for privacy-sensitive data, while for non-private data, retraining remains computationally expensive due to large-scale models and datasets. We therefore maintain a small replay buffer $D_r \subset D_r^{full}$ such that $|D_r| \ll |D_r^{full}|$ (at least 10% of $|D_r^{full}|$), here $D_r^{full}$ represents full retaining data [1]. The consequences of insufficient or absent buffer size are discussed in detail in Section 6.1.

Before adaptation, $\mathcal{M}$ performs well on $D_r$ and $D_f$ but poorly on $D_{\text{new}}$, i.e.,

$$f_{\mathcal{M}} : \mathcal{X}_{D_f} \rightarrow \mathcal{Y}_{D_f} \; ; \; \mathcal{X}_{D_r} \rightarrow \mathcal{Y}_{D_r} \; ; \; \mathcal{X}_{D_{\text{new}}} \nrightarrow \mathcal{Y}_{D_{\text{new}}}. \tag{1}$$

Let the adaptation algorithm be $\mathcal{F}$ for more details please refer Section 4 which modifies the model $\mathcal{M}$ to obtain $\mathcal{M}'$ by using $\mathcal{F}$ such that $\mathcal{M}' = \mathcal{F}(\mathcal{M}, D_f, D_r, D_{\text{new}})$ such that $\mathcal{M}'$ holds a new mapping relationship $f_{\mathcal{M}'}$ as

$$f_{\mathcal{M}'} : \mathcal{X}_{D_f} \nrightarrow \mathcal{Y}_{D_f} \; ; \; \mathcal{X}_{D_r} \rightarrow \mathcal{Y}_{D_r} \; ; \; \mathcal{X}_{D_{\text{new}}} \rightarrow \mathcal{Y}_{D_{\text{new}}}. \tag{2}$$

Here, $\nrightarrow$ denotes that the mapping no longer holds (i.e., the model has forgotten the corresponding mapping), while $\rightarrow$ indicates that the mapping still holds.

We now extend this problem to the continual setting, where the model is required to sequentially adopt new knowledge, involving both unlearning and learning across tasks $T$. These tasks may be triggered by requests from the user, the owner, or both. Here, $t$ is an index representing the task number, ranging from $0, 1, 2, \ldots, t, \ldots, T$. let $D_r^{(t)}$ be the data to be retained, $D_f^{(t)}$ be the data to be forgotten and $D_{\text{new}}^{(t)}$ be the data to be learned at $t_{\text{th}}$ step respectively.

The algorithm $\mathcal{F}$ is applied as:

$$\mathcal{M}^{(t)} = \mathcal{F}(\mathcal{M}^{(t-1)}, D_f^{(t)}, D_r^{(t)}, D_{\text{new}}^{(t)})$$

The adaptation algorithm $\mathcal{F}$ takes the previous-step model $\mathcal{M}^{(t-1)}$ along with the datasets $D_r^{(t)}$, $D_f^{(t)}$, and $D_{\text{new}}^{(t)}$ to produce the updated model $\mathcal{M}^{(t)}$. Thus, $\mathcal{F}$ handles adaptation requests sequentially, starting from $\mathcal{M}$ and generating a sequence of models $\mathcal{M}^{(1)}$, $\mathcal{M}^{(2)}$, $\mathcal{M}^{(3)}$, ..., $\mathcal{M}^{(t)}$, ..., $\mathcal{M}^{(T)}$, where $\mathcal{M}^{(t)}$ represents the modified model after the $t$-th adaptation task.

After step $t$, the adapted model $\mathcal{M}^{(t)}$ must satisfy:

---

[1] From this point, $D_r$ denotes the experience replay buffer unless specified.

$$f_{\mathcal{M}^{(t)}} : \mathcal{X}_{D_f^{(i)}} \not\rightarrow \mathcal{Y}_{D_f^{(i)}}, \quad \forall i \leq t \tag{3a}$$

$$f_{\mathcal{M}^{(t)}} : \mathcal{X}_{D_r^{(t)}} \rightarrow \mathcal{Y}_{D_r^{(t)}} \tag{3b}$$

$$f_{\mathcal{M}^{(t)}} : \mathcal{X}_{D_{\text{new}}^{(i)}} \rightarrow \mathcal{Y}_{D_{\text{new}}^{(i)}} \tag{3c}$$

We define successful CLU as

- Forgetting: $\text{Acc}(\mathcal{M}^{(t)}, D_f^{(i)}) \leq \frac{1}{C}, \quad \forall i \leq t$

- Retention: $\text{Acc}(\mathcal{M}^{(t)}, D_r^{(t)}) \approx \text{Acc}(\text{Oracle}, D_r^{(t)})$

- Learning: $\text{Acc}(\mathcal{M}^{(t)}, D_{\text{new}}^{(t)}) \approx \text{Acc}(\text{Oracle}, D_{\text{new}}^{(t)})$

where $C$ is the number of classes and Oracle denotes a model trained directly on target classes only.

### 3.2 Assumptions

BID-LoRA assumes the following conditions hold:

1. A replay buffer $D_r$ with $|D_r| \geq 0.1|D_r^{full}|$ is available.

2. Learning and unlearning occur simultaneously; if only one is needed, the corresponding loss is masked (see Section 4.3).

3. Data partitions satisfy $D = D_r^{full} \cup D_f$, $D_r^{full} \cap D_f = \emptyset$, $D_{\text{new}} \cap D = \emptyset$.

## 4 Method

### 4.1 Overview

Given the CLU objectives in Section 1, we propose Bi-Directional LoRA (BID-LoRA). The name reflects the framework dual nature: one direction incorporates new knowledge Eq. 3c, while the other removes unwanted knowledge Eq. 3a, all while preserving retained knowledge Eq. 3b.

BID-LoRA employs three dedicated LoRA adapters each with a separate pathway for retention, acquisition, and forgetting along with corresponding loss functions. This pathway separation prevents gradient interference between competing objectives. To maintain parameter efficiency, only lightweight adapters in attention layers are trained while the backbone remains frozen, following evidence that smaller network alterations reduce catastrophic forgetting Zhao et al. (2024). Additionally, a replay buffer $D_r$ (see Section 3) mitigates catastrophic forgetting of retained classes. Section 4.3 details the loss functions.

### 4.2 BID-LoRA

*LoRA-based Model Tuning:* We employ LoRA-based fine-tuning on attention layers as shown in Fig 2, which encode significant knowledge in transformer architectures. A standard linear layer computes $h = Wx$, where $x \in \mathbb{R}^k$ is the input vector, $h \in \mathbb{R}^d$ is the output vector, and $W \in \mathbb{R}^{d \times k}$ is the weight matrix. LoRA introduces a low-rank update $\Delta W = BA$, where $B \in \mathbb{R}^{d \times r}$ and $A \in \mathbb{R}^{r \times k}$ are low-rank matrices, yielding $h = Wx + \mathcal{S}(BAx)$ with scaling factor $\mathcal{S}$.

However, standard LoRA in CLU settings suffers from knowledge leakage gradual degradation of retained knowledge across successive steps (see Section 6.2). We hypothesize this occurs because a single adapter conflates three competing objectives: retention, acquisition, and forgetting. Interference among these objectives causes cumulative drift resulting in knowledge leakage.

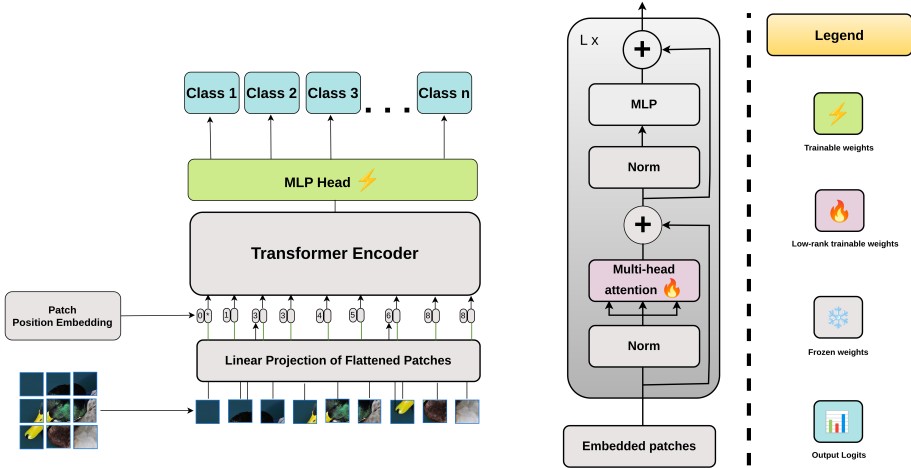

Figure 2: LoRA placement in BID-LoRA at Attention Modules

*Pathway Separation:* To address cumulative knowledge drift, i.e., knowledge leakage, we dedicate distinct adapters to each objective: $W_f = B_f A_f$ for forgetting, $W_{ret} = B_{ret} A_{ret}$ for retention, and $W_{new} = B_{new} A_{new}$ for learning new classes. Critically, $W_f$ learns weights that *nullify* forget-class knowledge, $W_{new}$ acquires new knowledge, while $W_{ret}$ preserves retained knowledge to counteract the knowledge leakage caused by both forget and new adapters. After sufficient training, adapters merge into the base weights $W$ as shwon in Fig 3

$$h = Wx + \mathcal{S}\left(B_{ret} A_{ret} + B_{new} A_{new} + B_f A_f\right)x \tag{1}$$

where $W \in \mathbb{R}^{d \times k}$ is frozen, and adapter consists of matrices $B_f, B_{ret}, B_{new} \in \mathbb{R}^{d \times r}$ and $A_f, A_{ret}, A_{new} \in \mathbb{R}^{r \times k}$ that serve their designated function. This separation provides two benefits: (1) drift in one adapter does not corrupt others, and (2) each adapter trains on its specific objective without gradient interference. Adapters merge at inference, preserving efficiency. We empirically validate pathway specialization in Section 5.4.4.

## 4.3  Loss Function

In this section, we present the loss functions which are designed to balance forgetting, retention, and new knowledge acquisition in our continual adaptation framework.

*Selective Forget Loss:* Existing unlearning methods suffer from strong unlearning signals that degrade retention knowledge, resulting in improper forgetting. To address this, we propose *Escape Unlearning*, a passive approach that pushes forget-class embeddings to an "escape" point maximally distant from all retain-class centroids, creating irreversible information loss. First, we compute the centroid of each class as:

$$c_k = \frac{1}{|D_k|} \sum_{x \in D_k} \text{emb}(x) \tag{2}$$

This yields retain centroids $\{c_{r_1}, c_{r_2}, \ldots, c_{r_K}\}$ and forget centroids $\{c_{f_1}, c_{f_2}, \ldots, c_{f_M}\}$. We obtain the escape direction by solving the minimax problem:

$$\mathbf{d}^* = \arg \min_{\|\mathbf{d}\|=1} \max_i \left(\mathbf{d}^\top c_{r_i}\right) \tag{3}$$

where $\mathbf{d}^* \in \mathbb{R}^{dim}$ is the escape direction vector in the *dim*-dimensional embedding space, $\mathbf{d}^\top$ denotes the transpose, and $\mathbf{d}^\top c_{r_i}$ computes the dot product measuring the projection of retain centroid $c_{r_i}$ along direction

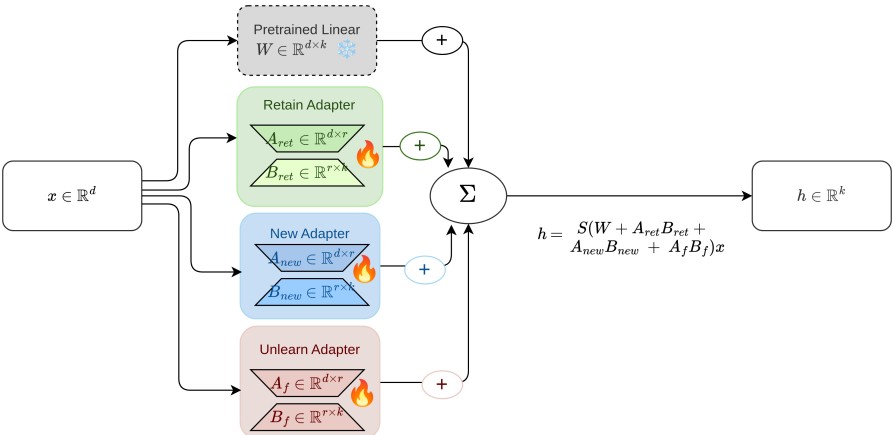

Figure 3: Pathway separation for BID-LoRA.

$\mathbf{d}$. The constraint $\|\mathbf{d}\| = 1$ ensures unit norm. The inner $\max_i$ identifies the retain centroid most aligned with $\mathbf{d}$, while the outer min searches over all unit directions to find $\mathbf{d}^*$ that minimizes this maximum alignment. Starting from a randomly initialized direction, the optimization iteratively converges to $\mathbf{d}^*$, yielding the direction maximally distant from all retain centroids.

However, placing the escape point on the unit sphere leads to unstable forgetting (see Section 5.4.6). Therefore, we scale the escape point away from the sphere:

$$\mathbf{t}_{\text{escape}} = \lambda_{\text{esc}} \cdot \mathbf{d}^* \tag{4}$$

This provides a target location away from retain embeddings in both direction and distance. The forget loss is then:

$$\mathcal{L}_f = \text{MSE}\left(\text{emb}(\mathbf{X}^f), \mathbf{t}_{\text{escape}}\right) \tag{5}$$

Where MSE indicates mean sqaured loss while this $\mathcal{L}_f$ pushes all forget sample embeddings toward the escape point, creating a many-to-one mapping that destroys class-discriminative information.

*Retention Loss* The retention loss preserves knowledge of retain classes and prevents model drift toward the escape point through two complementary terms:

$$\mathcal{L}_{ret} = \lambda_{\text{ce}} \cdot \text{CE}(\mathbf{z}_r, y_r) + \lambda_{\text{emb}} \cdot \text{MSE}(\mathbf{e}_r, \mathbf{e}_t) \tag{6}$$

where $\mathbf{z}_r$ are student logits for retain samples, $y_r$ are ground truth labels, and $\mathbf{e}_r$, $\mathbf{e}_t$ are student and teacher embeddings respectively. The embedding anchor term $\text{MSE}(\mathbf{e}_r, \mathbf{e}_t)$ prevents representation drift by keeping student embeddings close to the teacher (i.e., away from escape point $\mathbf{d}^*$), where the teacher is a frozen copy of the model at initialization.

*New Knowledge Loss* The model must simultaneously learn new classes using standard cross-entropy:

$$\mathcal{L}_{new} = \text{CE}(\mathbf{z}_n, y_n) \tag{7}$$

where CE indicates cross entropy loss $\mathbf{z}_n$ are logits and $y_n$ are labels for new class samples.

Each loss backpropagates exclusively through its designated adapter and classifier head nodes. During gradient updates, non-target adapters are frozen and non-target head gradients are masked to zero. Specifically, $\mathcal{L}_f$, $\mathcal{L}_{ret}$, and $\mathcal{L}_{new}$ update only their respective adapters $(B, A)$ and corresponding classifier head nodes.

This ensures forgetting cannot corrupt retained knowledge and each adapter specializes to its function, as per assumption 2 (Section 3.2), if only learning or unlearning is required, the corresponding loss terms are masked accordingly.

Algorithm 1 summarizes the complete BID-LoRA training procedure. Each iteration performs gradient-isolated updates for retain, forget, and new pathways sequentially, ensuring no cross-pathway interference. After training, adapters merge into base weights for efficient inference.

---

**Algorithm 1** BID-LoRA Training

---

**Require:** Pre-trained model $\mathcal{M}$, retain data $D_r^{full}$, forget data $D_f$, new data $D_{new}$, reply buffers be $D_r$ and $\mathcal{S}$ be the LoRA scaling factor
**Ensure:** Updated model with merged adapters
1: Initialize adapters $(B_{ret}, A_{ret})$, $(B_{new}, A_{new})$, $(B_f, A_f)$
2: Freeze backbone weights $W$
3: $\mathcal{M}_t \leftarrow \text{copy}(\mathcal{M})$                   ▷ Frozen teacher for embedding anchor
4: **// Compute escape point**
5: **for** each class $k$ in $D_r$ **do**
6:      $c_k \leftarrow \frac{1}{|D_k|} \sum_{x \in D_k} \text{emb}(x)$                 ▷ Retain centroids
7: **end for**
8: $\mathbf{d}^* \leftarrow \arg\min_{\|\mathbf{d}\|=1} \max_i (\mathbf{d}^\top c_{r_i})$          ▷ Escape direction
9: $\mathbf{t}_{\text{escape}} \leftarrow \lambda_{\text{esc}} \cdot \mathbf{d}^*$                 ▷ Escape point
10: **for** epoch $= 1$ to $E$ **do**
11:      **for** each mini-batch $(X_{ret}, y_{ret}), (X_f, y_f), (X_{new}, y_{new})$ from $(D_r, D_f, D_{new})$ **do**
12:          **// Retention update**
13:          Freeze $(B_f, A_f)$, $(B_{new}, A_{new})$
14:          $\mathbf{z}_{ret} \leftarrow \mathcal{M}(X_{ret})$               ▷ Student logits
15:          $\mathbf{e}_{ret} \leftarrow \text{emb}(X_{ret})$             ▷ Student embeddings
16:          $\mathbf{e}_{ret,t} \leftarrow \mathcal{M}_t.\text{emb}(X_{ret})$        ▷ Teacher embeddings
17:          $\mathcal{L}_{\text{ret}} \leftarrow \lambda_{\text{ce}} \cdot \text{CE}(\mathbf{z}_{ret}, y_{ret}) + \lambda_{\text{emb}} \cdot \text{MSE}(\mathbf{e}_{ret}, \mathbf{e}_{t,ret})$
18:          Update $(B_{\text{ret}}, A_{\text{ret}})$ and retain-class head
19:          **// Forget update**
20:          Freeze $(B_{ret}, A_{ret})$, $(B_{new}, A_{new})$
21:          $\mathcal{L}_{\text{forget}} \leftarrow \text{MSE}(\text{emb}(X_f), \mathbf{t}_{\text{escape}})$
22:          Update $(B_f, A_f)$ and forget-class head
23:          **// New knowledge update**
24:          Freeze $(B_{ret}, A_{ret})$, $(B_f, A_f)$
25:          $\mathcal{L}_{new} \leftarrow \text{CE}(\mathcal{M}(X_{new}), y_n)$
26:          Update $(B_{new}, A_{new})$ and new-class head
27:      **end for**
28: **end for**
29: **// Merge adapters**
30: $W \leftarrow W + \mathcal{S}(B_{ret}A_{ret} + B_{new}A_{new} - B_f A_f)$
31: **return** $\mathcal{M}$

---

# 5 Experimental

## 5.1 Experimental Setup

*Datasets and Pre-trained Models:* We evaluate BID-LoRA on classification and face recognition tasks. For classification, we use CIFAR-100 Krizhevsky et al. (2009) and adopt data-efficient image transformers Touvron et al. (2021) as the backbone. For face recognition, we use CASIA-Face100, which contains 100 identities sampled from CASIA-WebFace Yi et al. (2014) and constructed in Zhao et al. (2024), with a Face Trans-

Table 1: Performance comparison across all continual learning-unlearning tasks on CIFAR-100.

| Method | Tunable ↓ | Task-1 | | | | | | Task-2 | | | | | |
|---|---|---|---|---|---|---|---|---|---|---|---|---|---|
| | | $Acc_f$ ↓ | $Acc_r$ ↑ | $Acc_n$ ↑ | $Acc_o$ ↑ | KL ↓ | MIA ≈ 0.5 | $Acc_f$ ↓ | $Acc_r$ ↑ | $Acc_n$ ↑ | $Acc_o$ ↑ | KL ↓ | MIA ≈ 0.5 |
| Oracle | 100% | – | – | – | 78.89 | – | – | – | – | – | 75.60 | – | – |
| LSF Shibata et al. (2021) | 100% | 0.00 | 70.21 | 80.23 | 73.55 | 0.13 | 0.63 | 0.00 | 70.31 | 78.27 | 72.96 | 0.89 | 0.54 |
| CLPU-DER++ Liu et al. (2022) | 100% | 0.27 | 70.30 | 70.27 | 70.29 | 1.47 | 0.58 | 0.00 | 68.32 | 73.47 | 70.04 | 0.79 | 0.57 |
| UniCLUN Chatterjee et al. (2024) | 100% | 2.37 | 72.39 | 75.32 | 73.37 | 1.27 | 0.64 | 0.20 | 67.27 | 74.63 | 69.72 | 1.56 | 0.52 |
| UG-CLU Huang et al. (2025) | 100% | 1.31 | 76.73 | 73.27 | 75.58 | 0.73 | 0.59 | 0.37 | 70.32 | 71.47 | 70.70 | 1.89 | 0.59 |
| UnCLe Adhikari et al. (2025) | 100% | 1.47 | 78.43 | 75.23 | 76.83 | 0.41 | 0.60 | 0.50 | 70.00 | 76.27 | 72.09 | 1.00 | 0.61 |
| **BID-LoRA** | 5.08% | 0.93 | 75.71 | 76.93 | 76.03 | 0.67 | 0.57 | 0.27 | 70.83 | 79.87 | 73.84 | 0.60 | 0.51 |

| Method | Tunable ↓ | Task-3 | | | | | | Task-4 | | | | | |
|---|---|---|---|---|---|---|---|---|---|---|---|---|---|
| | | $Acc_f$ ↓ | $Acc_r$ ↑ | $Acc_n$ ↑ | $Acc_o$ ↑ | KL ↓ | MIA ≈ 0.5 | $Acc_f$ ↓ | $Acc_r$ ↑ | $Acc_n$ ↑ | $Acc_o$ ↑ | KL ↓ | MIA ≈ 0.5 |
| Oracle | 100% | – | – | – | 77.64 | – | – | – | – | – | 78.04 | – | – |
| LSF Shibata et al. (2021) | 100% | 1.00 | 72.00 | 77.25 | 73.75 | 0.89 | 0.53 | 0.00 | 75.00 | 73.35 | 74.45 | 0.78 | 0.52 |
| CLPU-DER++ Liu et al. (2022) | 100% | 0.48 | 71.33 | 80.25 | 74.31 | 1.35 | 0.58 | 0.25 | 79.28 | 72.25 | 76.94 | 0.62 | 0.57 |
| UniCLUN Chatterjee et al. (2024) | 100% | 0.36 | 68.48 | 79.00 | 71.99 | 2.58 | 0.61 | 0.36 | 73.25 | 70.89 | 72.46 | 1.85 | 0.61 |
| UG-CLU Huang et al. (2025) | 100% | 0.00 | 74.58 | 78.25 | 75.81 | 0.78 | 0.59 | 0.00 | 70.25 | 74.89 | 71.80 | 1.87 | 0.63 |
| UnCLe Adhikari et al. (2025) | 100% | 0.00 | 71.00 | 81.25 | 74.42 | 0.77 | 0.55 | 0.89 | 74.36 | 71.58 | 73.43 | 0.88 | 0.51 |
| **BID-LoRA** | 5.08% | 0.13 | 72.33 | 79.73 | 74.80 | 0.76 | 0.54 | 0.27 | 76.00 | 78.13 | 76.71 | 0.69 | 0.50 |

| Method | Tunable ↓ | Task-5 | | | | | | Task-6 | | | | | |
|---|---|---|---|---|---|---|---|---|---|---|---|---|---|
| | | $Acc_f$ ↓ | $Acc_r$ ↑ | $Acc_n$ ↑ | $Acc_o$ ↑ | KL ↓ | MIA ≈ 0.5 | $Acc_f$ ↓ | $Acc_r$ ↑ | $Acc_n$ ↑ | $Acc_o$ ↑ | KL ↓ | MIA ≈ 0.5 |
| Oracle | 100% | – | – | – | 78.44 | – | – | – | – | – | 79.16 | – | – |
| LSF Shibata et al. (2021) | 100% | 0.00 | 68.98 | 78.91 | 73.75 | 1.25 | 0.53 | 0.00 | 65.00 | 75.37 | 68.46 | 1.24 | 0.55 |
| CLPU-DER++ Liu et al. (2022) | 100% | 0.00 | 69.00 | 75.89 | 74.31 | 1.78 | 0.59 | 0.87 | 67.81 | 79.87 | 71.83 | 1.36 | 0.57 |
| UniCLUN Chatterjee et al. (2024) | 100% | 0.22 | 73.25 | 74.25 | 71.99 | 1.89 | 0.61 | 0.00 | 68.50 | 83.00 | 73.33 | 0.98 | 0.67 |
| UG-CLU Huang et al. (2025) | 100% | 0.00 | 71.35 | 76.36 | 75.81 | 0.69 | 0.66 | 0.00 | 63.51 | 81.29 | 69.44 | 0.87 | 0.53 |
| UnCLe Adhikari et al. (2025) | 100% | 0.87 | 70.98 | 77.82 | 74.42 | 0.78 | 0.67 | 0.00 | 67.00 | 80.00 | 71.33 | 0.98 | 0.54 |
| **BID-LoRA** | 5.08% | 0.67 | 72.33 | 80.80 | 75.16 | 0.63 | 0.50 | 0.00 | 68.67 | 83.20 | 73.51 | 0.79 | 0.51 |

former Zhong & Deng (2021) as the backbone. We use $D_r$ as 10% of $D_r^{\text{full}}$ as buffer. We use uniform rank 8 for both retain and new adapters, while the forget adapter uses rank 4.

*Evaluation Protocol:* We implement a six-task evaluation protocol where the model progressively transitions from classes 0-29 to 60-89. Each task involves: (i) retaining 20 classes, (ii) forgetting 10 classes, and (iii) learning 10 new classes. Starting from a pre-trained checkpoint on classes 0-29, each subsequent task slides the class window by 10 positions: task 1 operates on classes 10-39, task 2 on classes 20-49, continuing until task 6 operates on classes 60-89 (Fig. 4). This sliding window design validates true continual adaptation through simultaneous learning and forgetting. Our protocol offers key advantages over existing methods. First, it tests longevity and sustainability claims often questionable under extended evaluation. Second, over the complete cycle, every piece of pretrained knowledge is systematically replaced, providing comprehensive assessment of adaptability. Third, this extensive protocol proves our approach's genuineness through sustained performance across multiple knowledge transitions rather than cherry-picked scenarios.

*Theoretical Best:* We compare all baseline methods and our approach against Oracle-model performance, which serves as the theoretical and practical upper bound. Oracle models are obtained by directly training on each target class range (e.g., classes 10-39 for task 1, classes 20-49 for task 2 ... utill task 6) without any low-rank adaptations, incremental learning, or unlearning approaches. This provides a clean baseline that follows the same 6-task sliding window protocol, allowing us to measure how close our continual adaptation approach comes to optimal performance. The Oracle comparison quantifies how close our continual adaptation approach comes to optimal retraining performance.

*Metrics:* We evaluate our approach based on the performance of forgotten, retained, and newly learned classes using class accuracy, Membership Inference Attack (MIA) success rate, and KL divergence. Following Huang

Table 2: Performance comparison across all continual learning-unlearning tasks on Face recognistion tasks on CASIA-Face100

| Method | Tunable ↓ | Task-1 | | | | | | Task-2 | | | | | |
|---|---|---|---|---|---|---|---|---|---|---|---|---|---|
| | | $Acc_f$ ↓ | $Acc_r$ ↑ | $Acc_n$ ↑ | $Acc_o$ ↑ | KL ↓ | MIA ≈ 0.5 | $Acc_f$ ↓ | $Acc_r$ ↑ | $Acc_n$ ↑ | $Acc_o$ ↑ | KL ↓ | MIA ≈ 0.5 |
| Oracle | 100% | – | – | – | 95.65 | – | – | – | – | – | 96.36 | – | – |
| LSF Shibata et al. (2021) | 100% | 0.00 | 90.87 | 93.51 | 91.75 | 1.27 | 0.57 | 0.00 | 87.98 | 90.73 | 88.90 | 1.58 | 0.51 |
| CLPU-DER++ Liu et al. (2022) | 100% | 0.58 | 88.98 | 90.36 | 89.44 | 1.58 | 0.61 | 0.32 | 88.96 | 93.54 | 90.49 | 2.50 | 0.50 |
| UniCLUN Chatterjee et al. (2024) | 100% | 0.87 | 90.28 | 90.96 | 90.51 | 1.00 | 0.62 | 0.89 | 90.70 | 90.35 | 90.58 | 0.87 | 0.57 |
| UG-CLU Huang et al. (2025) | 100% | 1.00 | 91.35 | 91.18 | 91.29 | 0.87 | 0.67 | 0.76 | 91.13 | 87.00 | 89.75 | 0.98 | 0.67 |
| UnCLe Adhikari et al. (2025) | 100% | 0.00 | 92.00 | 94.83 | 92.94 | 2.25 | 0.61 | 0.00 | 90.25 | 91.27 | 90.59 | 0.78 | 0.61 |
| **BID-LoRA** | 5.00% | 0.10 | 91.93 | 95.72 | 93.20 | 0.98 | 0.57 | 0.88 | 91.36 | 93.55 | 92.09 | 1.10 | 0.53 |

| Method | Tunable ↓ | Task-3 | | | | | | Task-4 | | | | | |
|---|---|---|---|---|---|---|---|---|---|---|---|---|---|
| | | $Acc_f$ ↓ | $Acc_r$ ↑ | $Acc_n$ ↑ | $Acc_o$ ↑ | KL ↓ | MIA ≈ 0.5 | $Acc_f$ ↓ | $Acc_r$ ↑ | $Acc_n$ ↑ | $Acc_o$ ↑ | KL ↓ | MIA ≈ 0.5 |
| Oracle | 100% | – | – | – | 94.85 | – | – | – | – | – | 96.87 | – | – |
| LSF Shibata et al. (2021) | 100% | 0.25 | 91.13 | 90.35 | 90.87 | 2.36 | 0.56 | 0.00 | 91.87 | 94.00 | 92.58 | 1.36 | 0.58 |
| CLPU-DER++ Liu et al. (2022) | 100% | 0.00 | 90.68 | 90.58 | 90.65 | 1.35 | 0.57 | 0.00 | 92.58 | 93.68 | 92.95 | 2.25 | 0.51 |
| UniCLUN Chatterjee et al. (2024) | 100% | 0.00 | 91.58 | 91.37 | 91.51 | 1.89 | 0.51 | 0.29 | 92.87 | 94.78 | 93.51 | 1.69 | 0.67 |
| UG-CLU Huang et al. (2025) | 100% | 0.00 | 89.69 | 92.35 | 90.58 | 0.78 | 0.51 | 0.78 | 91.25 | 92.65 | 91.72 | 1.87 | 0.64 |
| UnCLe Adhikari et al. (2025) | 100% | 0.00 | 92.22 | 93.00 | 92.48 | 0.91 | 0.61 | 0.10 | 90.36 | 92.82 | 91.18 | 0.94 | 0.53 |
| **BID-LoRA** | 5.00% | 0.00 | 92.25 | 92.35 | 92.29 | 0.78 | 0.54 | 0.00 | 92.51 | 95.20 | 93.40 | 0.87 | 0.57 |

| Method | Tunable ↓ | Task-5 | | | | | | Task-6 | | | | | |
|---|---|---|---|---|---|---|---|---|---|---|---|---|---|
| | | $Acc_f$ ↓ | $Acc_r$ ↑ | $Acc_n$ ↑ | $Acc_o$ ↑ | KL ↓ | MIA ≈ 0.5 | $Acc_f$ ↓ | $Acc_r$ ↑ | $Acc_n$ ↑ | $Acc_o$ ↑ | KL ↓ | MIA ≈ 0.5 |
| Oracle | 100% | – | – | – | 94.58 | – | – | – | – | – | 95.85 | – | – |
| LSF Shibata et al. (2021) | 100% | 0.00 | 89.69 | 91.87 | 90.42 | 1.36 | 0.61 | 0.00 | 90.87 | 91.27 | 91.00 | 1.25 | 0.57 |
| CLPU-DER++ Liu et al. (2022) | 100% | 0.00 | 87.25 | 92.35 | 88.95 | 2.58 | 0.68 | 0.00 | 90.81 | 91.28 | 90.97 | 1.57 | 0.51 |
| UniCLUN Chatterjee et al. (2024) | 100% | 0.25 | 87.25 | 93.15 | 89.22 | 1.24 | 0.57 | 0.00 | 87.65 | 90.87 | 88.72 | 0.87 | 0.59 |
| UG-CLU Huang et al. (2025) | 100% | 0.36 | 89.69 | 93.00 | 90.79 | 0.98 | 0.51 | 0.00 | 88.61 | 90.66 | 89.29 | 0.98 | 0.67 |
| UnCLe Adhikari et al. (2025) | 100% | 0.00 | 90.11 | 92.55 | 90.92 | 0.87 | 0.53 | 0.00 | 89.95 | 91.69 | 90.53 | 1.18 | 0.60 |
| **BID-LoRA** | 5.00% | 0.00 | 92.30 | 93.25 | 92.62 | 0.74 | 0.51 | 0.00 | 90.98 | 91.70 | 91.22 | 0.77 | 0.50 |

et al. Huang et al. (2025), we define: forget accuracy $Acc_f$ as performance on forgotten data, retain accuracy $Acc_r$ as performance on retained data, new accuracy $Acc_n$ as performance on newly added data, and overall accuracy $Acc_o$ as performance across combined retained and new classes. To verify effective unlearning, we measure the MIA success rate as suggested by Choquette-Choo et al. (2021), and compute KL divergence between each model and the oracle model to assess logit-level similarity. We also report the tunable ratio, defined as the proportion of parameters updated during CLU steps, to indicate practical usability under resource-constrained scenarios.

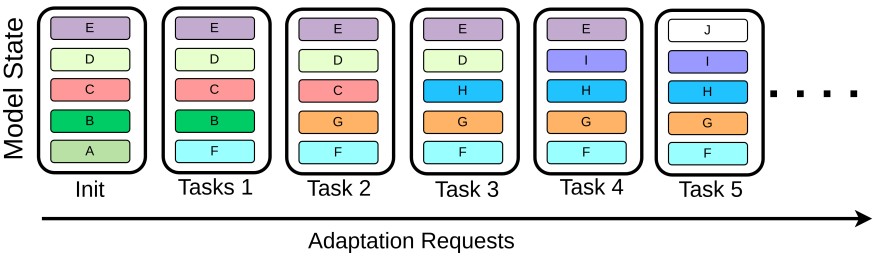

Figure 4: Illustration of continual adapting evaluation protocol.

Ideally, $Acc_f$ should approach zero, while $Acc_r$, $Acc_n$, and $Acc_o$ should align with the oracle model's performance. The tunable ratio should approach zero for parameter efficiency. The MIA success rate should

approximate 0.5, indicating the adversary cannot distinguish between unlearned and never-seen knowledge. KL divergence should approach zero, indicating no difference at the logit level between the oracle and adapted models.

## 5.2 Baseline Implementations

For baselines, we adopt implementations of existing CLU methods discussed in Section 2.4. These include: LSF by Shibata et al. Shibata et al. (2021), which introduced the CLU problem and provided an initial solution; CLPU-DER++ by Liu et al. Liu et al. (2022), which employs temporal networks for cleaner removal of forget knowledge; UniCLUN by Chatterjee et al. Chatterjee et al. (2024), a student-teacher distillation approach; UnCLe by Adhikari et al. Adhikari et al. (2025), which explores hypernetworks for data-free unlearning; and UG-CLU by Huang et al. Huang et al. (2025), a weight saliency-based method.

## 5.3 Results Discussion

Tables 1 and 2 present results for CIFAR-100 classification and CASIA-Face100 recognition. BID-LoRA achieves first (green) or second-best (blue) performance across nearly all metrics while using only $\approx 5.08\%$ tunable parameters compared to 100% for all baselines.

*Unlearning Analysis:* BID-LoRA maintains forget accuracy between 0–0.93% on CIFAR-100 (e.g., 0.93% in Task 1, 0.27% in Task 2, 0.13% in Task 3) and 0–0.88% on face recognition, confirming effective unlearning. On CIFAR-100, BID-LoRA achieves the best or second-best overall accuracy in 5 of 6 tasks: 76.03% (Task 1), 73.84% (Task 2), 74.80% (Task 3), 76.71% (Task 4), 75.16% (Task 5), and 73.51% (Task 6). The MIA scores cluster around the ideal value of 0.5 (ranging 0.50–0.57), indicating successful privacy protection against membership inference attacks.

*Knowledge Leakage Analysis:* Knowledge leakage, measured by overall accuracy degradation across tasks, remains minimal for BID-LoRA. While baseline methods such as LSF Shibata et al. (2021) and UniCLUN Chatterjee et al. (2024) show cumulative accuracy drops of 3–8% from Task 1 to Task 6, BID-LoRA maintains stable performance with only 2.52% variation on CIFAR-100 (76.03% $\rightarrow$ 73.51%) and 1.98% on face recognition (93.20% $\rightarrow$ 91.22%). The KL divergence remains consistently low: 0.60–0.79 on CIFAR-100 and 0.74–1.10 on face recognition, demonstrating that BID-LoRA successfully minimizes knowledge leakage in extended CLU scenarios.

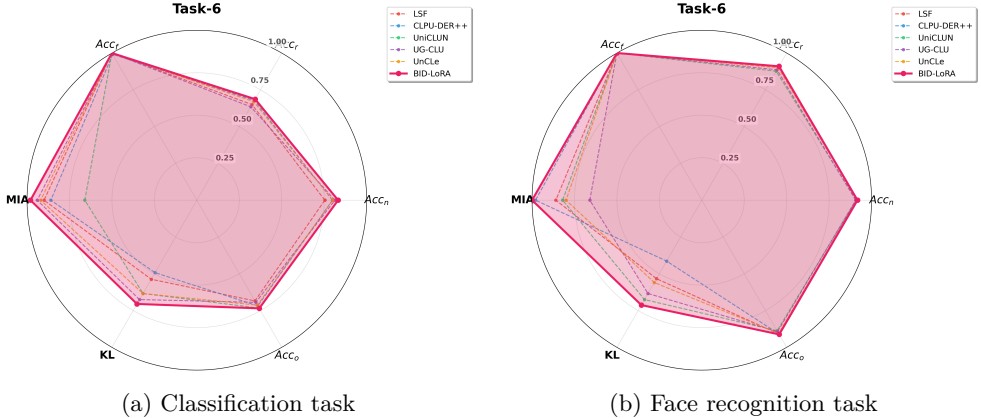

(a) Classification task        (b) Face recognition task

Figure 5: Radar plot comparison at Task-6. BID-LoRA consistently outperforms all baselines across metrics on both classification and face recognition tasks.

*Baseline Comparison:* Combined baseline methods exhibit conflicting optimization between their CL and MU components. LSF Shibata et al. (2021) achieves strong unlearning ($Acc_f = 0\%$ in most tasks) but

sacrifices overall accuracy, dropping to 68.46% in Task 6 on CIFAR-100. CLPU-DER++Liu et al. (2022) and UniCLUN Chatterjee et al. (2024) demonstrate unstable retention accuracy, with UniCLUN's $Acc_r$ falling to 67.27% by Task 2 on CIFAR-100. UG-CLU Huang et al. (2025) and UnCLe Adhikari et al. (2025) show inconsistent MIA scores reaching up to 0.67, indicating incomplete privacy protection. On face recognition, BID-LoRA consistently outperforms baselines, achieving $Acc_o$ of 93.20%, 92.09%, 92.29%, 93.40%, 92.62%, and 91.22% across Tasks 1–6 respectively, while baselines remain below 93% in most cases.As illustrated in Fig 5, BID-LoRA's radar plot at Task-6 encloses all baseline methods, demonstrating superior performance across all evaluation metrics on both classification and face recognition tasks. Similar patterns are observed across all tasks on CIFAR-100 and CASIA-Face100 benchmarks.

*Convergence Patterns:* BID-LoRA exhibits convergent behavior where new knowledge accuracy $Acc_n$ aligns closely with overall accuracy while retained accuracy ($Acc_r$) remains stable. On CIFAR-100, new knowledge accuracy ($Acc_n$) improves from 76.93% in Task 1 to 83.20% in Task 6, while overall accuracy $Acc_o$ remains stable (76.03% to 73.51%), demonstrating effective knowledge acquisition without degrading cumulative performance. Retained accuracy maintains consistency across Tasks 2–6 (70.83%–76.00%), demonstrating that the bidirectional adapter architecture effectively balances learning and unlearning without catastrophic forgetting.

### 5.4 Ablation Study

#### 5.4.1 Parameter Efficiency

We study how LoRA rank affects performance on DeiT-Tiny over 10 epochs. Table 3 shows performance plateaus beyond rank 8, achieving effective adaptation with only ≈5% of parameters versus nearly 100% for existing methods Shibata et al. (2021); Adhikari et al. (2025); Huang et al. (2025); Chatterjee et al. (2024); Liu et al. (2022).

Table 3: Ablation study on the rank of LoRA modules

| Rank | % Ratio | $Acc_f$ | $Acc_r$ | $Acc_n$ | $Acc_o$ |
|---|---|---|---|---|---|
| 1 | 1.09 | 1.73 | 48.57 | 27.60 | 41.58 |
| 2 | 2.30 | 3.47 | 46.43 | 43.07 | 45.31 |
| 4 | 3.24 | 0.00 | 65.71 | 41.87 | 57.77 |
| 8 | 5.08 | 0.80 | 76.43 | 69.60 | 74.15 |
| 16 | 8.56 | 1.07 | 81.43 | 73.73 | 78.86 |
| 32 | 14.80 | 0.40 | 78.57 | 76.93 | 78.03 |
| 64 | 25.03 | 0.40 | 78.57 | 79.07 | 78.74 |

#### 5.4.2 BID-LoRA vs Standard LoRA

As discussed in Section 4, BID-LoRA employs distinct pathways to handle conflicting tasks like continual learning and unlearning simultaneously. Table 4 shows BID-LoRA outperforms standard LoRA (applied to QKV attention and classification head) in the CLU setting over 20 epochs, demonstrating that pathway separation minimizes knowledge leakage between conflicting objectives.

Table 4: BID-LoRA vs. Standard LoRA in the CLU setting

| Method | % Ratio | $Acc_f$ | $Acc_r$ | $Acc_n$ | $Acc_o$ |
|---|---|---|---|---|---|
| Standard LoRA | 2.54 | 0.00 | 74.29 | 85.07 | 77.88 |
| BID-LoRA | 5.08 | 0.13 | 77.14 | 85.87 | 80.05 |

### 5.4.3 Buffer Ratio

Buffer data informs the model what to retain or forget. Here, we examine the effect of retain buffer ratio on overall accuracy. Our method is designed to remain robust even with limited buffer availability, as shown in Table. 5.

Table 5: Ablation on retain ratio

| Retain Ratio | Speed | $Acc_f$ | $Acc_r$ | $Acc_n$ | $Acc_o$ |
|---|---|---|---|---|---|
| 1.0 | $1.0\times$ | 0.40 | 68.00 | 81.74 | 72.58 |
| 0.5 | $2.0\times$ | 0.00 | 70.00 | 71.05 | 70.35 |
| 0.3 | $3.3\times$ | 0.00 | 66.00 | 79.11 | 70.37 |
| 0.1 | $10\times$ | 0.00 | 65.00 | 79.96 | 69.98 |

### 5.4.4 Ablation on Adapters

Table 6: Ablation on adapter pathway contributions (F=Forget, R=Retain, N=New)

| F | R | N | $Acc_f$ | $Acc_r$ | $Acc_n$ | $Acc_o$ |
|---|---|---|---|---|---|---|
| Pre-train | | | 73.64 | 82.27 | 0.00 | 48.49 |
| ✗ | ✓ | ✓ | 70.45 | 82.73 | 51.82 | 72.43 |
| ✓ | ✗ | ✓ | 0.00 | 0.23 | 64.55 | 21.67 |
| ✓ | ✓ | ✓ | 0.00 | 84.09 | 56.82 | 75.00 |
| ✓ | ✓ | ✗ | 0.00 | 84.09 | 0.00 | 56.06 |

This experiment verifies that each adapter pathway holds knowledge specific to its task. By selectively disabling pathways, we observe accuracy drops for retain and new tasks, while forget accuracy increases when its pathway is disabled. As shown in Table 6, the optimal performance is achieved only when all three pathways are active, validating the necessity of the tri-pathway architecture. The original classification head is restored during evaluation to isolate adapter contributions. Results are obtained over 15 epochs with pathways disabled during testing.

### 5.4.5 Geometric Verification of Unlearning

A critical question is whether forget-class embeddings actually move toward the computed escape direction $d^*$. Fig 6(a,b) shows t-SNE visualizations before and after unlearning. Initially, the three forget classes (0, 1, 2) are spatially separated from $d^*$. After unlearning, all three forget clusters collapse toward the dustbin point, demonstrating successful alignment with the intended escape direction. The 3D sphere visualizations (Fig 6 (c,d)) provide additional evidence by showing class centroids projected onto a unit sphere via PCA. While retain-class centroids (R3–R9, squares) remain anchored at their original positions relative to the retain centroid $\bar{c}_r$, the forget centroids (F0, F1, F2, circles) migrate along the antipodal axis toward $d^*$. Notably, F1 moves from a position distant from $d^*$ to near-perfect alignment after unlearning, while F0 and F2 also show substantial movement toward the escape direction. This confirms that the optimization successfully drives forget-class representations away from retain classes along the computed global escape vector.

### 5.4.6 Escape Point Scaling

As discussed in Section 4.3, placing $d^*$ on the unit sphere leads to unstable forgetting due to nearby centroids. tablele 7 validates this: $\lambda_{esc} = 10$ achieves optimal forgetting (0.53%) by pushing the escape point beyond the embedding sphere.

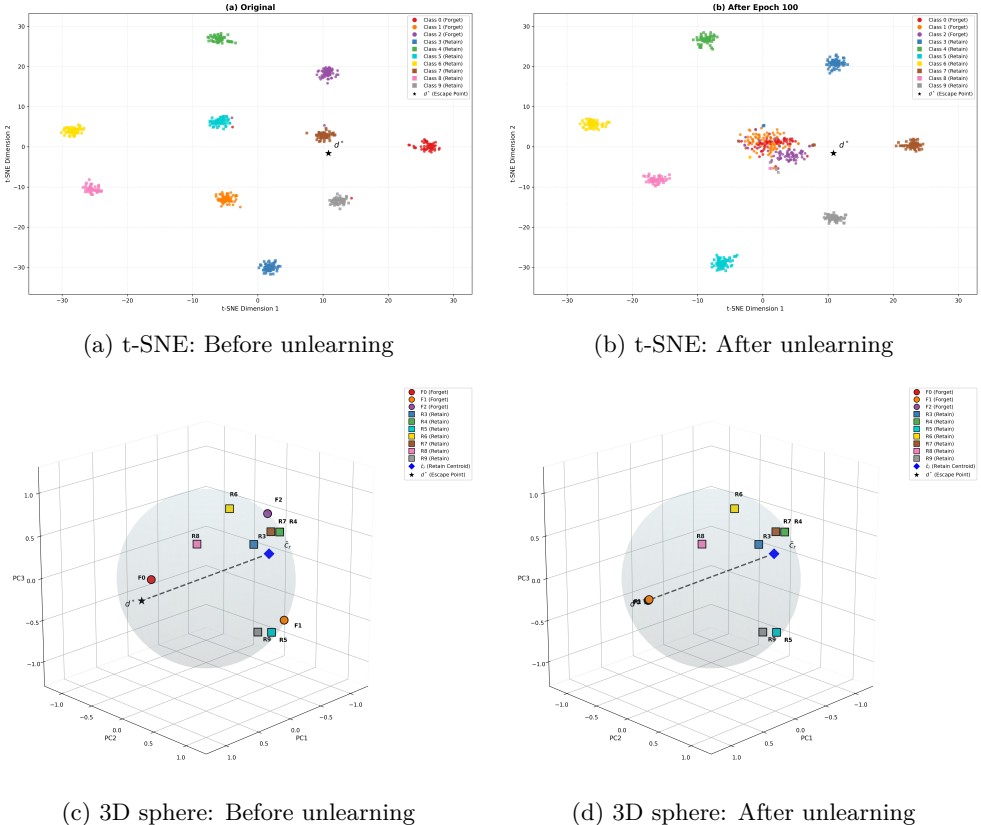

(a) t-SNE: Before unlearning        (b) t-SNE: After unlearning

(c) 3D sphere: Before unlearning        (d) 3D sphere: After unlearning

Figure 6: Geometric verification of unlearning. Top row: t-SNE visualization showing forget classes migrating toward escape point $d^*$. Bottom row: 3D hypersphere visualization with dashed antipodal axis from $\bar{c}_r$ to $d^*$.

# 6 Discussion

In this section, we provide an in-depth discussion of several critical aspects of our approach. We examine the theoretical necessity of retain data in machine unlearning, present an algorithmic perspective of our BID-LoRA method, and analyze the knowledge leakage phenomenon that motivates our unified framework. These discussions provide deeper insights into the design principles and empirical observations underlying our work.

## 6.1 On the Necessity of Retain Buffer in Machine Unlearning

Effective unlearning requires distinguishing what to forget from what to retain. This section investigates whether forget data ($D_f$) or retain buffer ($D_r$, a subset of full retain data $D_r^{\text{full}}$) can be avoided in classification and face recognition tasks.

We find $D_r$ is fundamentally unavoidable for preserving task-critical knowledge. Without architectural support (e.g., modular pathways), all parameters are adjusted using both $D_f$ and $D_r^{\text{full}}$ during pretraining, making disentanglement infeasible without $D_r$ access.

Only noise-based impair-and-repair methods Tarun et al. (2023) and source-free unlearning Ahmed et al. (2025) attempt to bypass data dependencies. The former still requires $D_r$. Source-free methods estimate retain data Hessians using only $D_f$ through semi-definite programming, but are limited to convex losses and linear classifiers, making them inapplicable to modern non-convex architectures like transformers and ResNets. In practice, they use frozen pre-trained feature extractors and unlearn only linear heads. While error bounds improve with dimensionality, Hessian storage and inversion become computationally prohibitive.

Table 7: Ablation on escape scaling factor $\lambda_{\text{esc}}$. Higher scaling pushes forget embeddings further from retain centroids, improving unlearning

| $\lambda_{\text{esc}}$ | $Acc_f \downarrow$ | $Acc_r \uparrow$ | $Acc_n \uparrow$ | $Acc_o \uparrow$ |
|---|---|---|---|---|
| 0 | 8.80 | 78.57 | 61.07 | 72.74 |
| 2 | 4.13 | 80.71 | 63.60 | 75.01 |
| 5 | 4.27 | 77.14 | 68.27 | 74.18 |
| 10 | 0.53 | 77.14 | 68.00 | 73.45 |

All established methods for non-convex models critically rely on $D_r$. Weight-importance methods (EWC Kirkpatrick et al. (2017), SALUN Fan et al. (2023)) compute Fisher information or saliency from $D_r$ to protect retain-relevant weights. SCRUB Kurmanji et al. (2023) and GS-LoRA Zhao et al. (2024) explicitly leverage $D_r$ to safeguard retained knowledge. CLU methods universally depend on $D_r$: LSF Shibata et al. (2021) uses mnemonic codes, UnCLe Adhikari et al. (2025) employs task embeddings, CLPU-DER++ Liu et al. (2022) maintains experience replay, and UniCLUN Chatterjee et al. (2024) and UG-CLU Huang et al. (2025) incorporate $D_r$ for stability.

Generative tasks present exceptions: FLAT Wang et al. (2024) solves unlearning without $D_r$ through regularizers, while ESD Gandikota et al. (2023) avoids both $D_f$ and $D_r$ using teacher models to generate synthetic samples.

While $D_f$ can theoretically be avoided through noise-based proxies or source-free methods, no viable approach eliminates $D_r$ dependence for non-convex deep learning without compromising retention performance.

### 6.2 Knowledge Leakage Analysis

Can we solve CLU by combining existing CL and MU methods? We pair four combinations (ER-ACE Caccia et al. (2021) + GS Zhao et al. (2024) ), (DER++ Buzzega et al. (2020) + FU Tarun et al. (2023)), (EWC Kirkpatrick et al. (2017) + SalUn Fan et al. (2023)), and (ER-AML Caccia et al. (2021) + GS Zhao et al. (2024)) on data-efficient image transformers Touvron et al. (2021) and FaceTransformer Zhong & Deng (2021) (face recognition). As shown in Fig 7, all exhibit progressive knowledge leakage across CLU cycles, confirming that piecewise solutions fail and a unified framework is necessary.

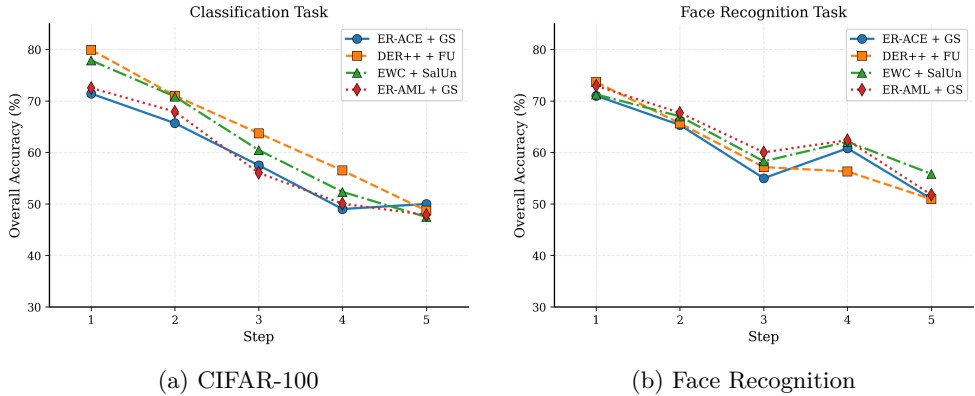

(a) CIFAR-100          (b) Face Recognition

Figure 7: Knowledge leakage in CL+MU combinations. Retain accuracy degrades progressively across CLU cycles on both benchmarks.

## 7 Conclusion

We introduced BID-LoRA, a parameter-efficient framework for Continual Learning and Unlearning (CLU) with three key contributions: (1) Escape Unlearning pushes forget-class embeddings to a scaled escape

point, enabling stable forgetting without disrupting retained knowledge; (2) dedicated adapters for retention, acquisition, and forgetting eliminate gradient interference and knowledge leakage; (3) state-of-the-art CLU performance with only $\approx 5\%$ tunable parameters. Experiments on CIFAR-100 and CASIA-Face100 show BID-LoRA outperforms all baselines across retain, forget, and new accuracy metrics. Our sliding window protocol establishes the first benchmark for validating complete knowledge replacement over multiple CLU cycles, offering a practical solution for real-world applications under privacy and regulatory constraints. Future work will (1) eliminate retention buffers entirely (reducing from 10% to 0%) and (2) extend to other biometric modalities such as iris and fingerprint recognition where privacy-driven unlearning is critical.

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
