# OpenReview forum: "BID-LoRA: A Parameter-Efficient Framework for Continual Learning and Unlearning"
_TMLR — Under review for TMLR_

### Review · Reviewer_9mgp · 2026-07-09

**Summary Of Contributions:**

The paper makes two contributions. It proposes BID-LoRA, a parameter-efficient tri-pathway LoRA framework that separates retain, forget, and new-knowledge updates to reduce interference. Second, it introduces escape unlearning, which moves forget-class embeddings toward an escape point away from retained-class centroids, and evaluates the method on CIFAR-100 and CASIA-Face100 using a multi-step sliding-window CLU protocol.

**Audience:**

Yes

**Audience Explanation:**

The paper addresses a timely problem at the intersection of continual learning, machine unlearning, and parameter-efficient adaptation, which is relevant to researchers working on model updating, privacy-preserving learning, and efficient fine-tuning. The proposed tri-pathway LoRA design and escape-unlearning objective also provide a concrete approach for reducing interference between retaining, forgetting, and learning objectives. However, the interest may be somewhat specialized rather than broad, since the evaluation is limited to vision classification and face recognition, and the method’s implications for larger foundation models or generative models remain unclear.

**Claims And Evidence:**

No

**Claims Explanation:**

The claims are partially supported. The paper provides evidence on CIFAR-100 and CASIA-Face100, including multi-step CLU results, comparisons with several baselines, and useful ablations on LoRA rank, pathway separation, buffer size, and escape scaling. However, the broader claims about practical deployment and privacy robustness are less convincing because the evaluation is limited to vision classification/face recognition, relies on a 10% retain buffer, and lacks stronger unlearning tests such as relearning, model inversion, extraction, or adaptive attacks. Overall, the results support the method’s promise in the tested settings but not its claims of broader generality or robustness.

**Requested Changes:**

I suggest the authors make the following changes. First, clarify the technical novelty of BID-LoRA relative to standard LoRA-based continual learning/unlearning methods, especially why the three-pathway design provides a principled advantage beyond separating objectives into different adapters. Second, strengthen the unlearning evaluation with more robust tests, such as relearning attacks, model inversion attacks, extraction attacks, and adaptive membership inference attacks. Third, discuss the practical limitation of requiring a 10% retain replay buffer, especially in privacy-sensitive settings where storing retain data may be restricted. Finally, broaden the evaluation beyond CIFAR-100 and CASIA-Face100, or more clearly limit the claims to vision classification and face-recognition settings rather than general CLU or foundation-model scenarios.

---

### Review · Reviewer_KYhU · 2026-07-12

**Summary Of Contributions:**

This paper discovers CL and MU, and proposes a new problem setting called CLU with three key goals, which need to forget selected classes and learning new classes over a sequence of adaptation requests. The authors then propose BID-LoRA, which can separate LoRA pathways for retention, forgetting, and new-class acquisition. The authors use an “escape unlearning” objective that maps forgotten-class embeddings toward a point selected to be distant from retained-class centroids.

Strengths
- The problem definition is interesting, particularly when enrollment and deletion requests occur repeatedly.
- The results examine adapter components, LoRA ranks, buffer ratios, and escape-point scaling.
- The reported trainable-parameter ratio is smaller than that of the implemented baselines.

**Additional Comments:**

- Bi-Directional LoRA is not clear; I think this should be three adapters, not two or bi-case.
- The manuscript contains many grammatical and typographical errors, including “reply buffers,” “mean sqaured,” “shwon,” “tablele,” “recognistion,” and several incomplete sentences.
- Equation numbering restarts in Section 4 despite earlier equations.
- Figure 6: it uses t-SNE and PCA as geometric evidence, but these projections can distort distances.
- Table 5’s “Speed” appears to be calculated directly --- not measured wall-clock speed.

**Audience:**

Yes

**Audience Explanation:**

This is a timely and relevant problem at the intersection of continual learning, machine unlearning, and parameter-efficient adaptation.

**Claims And Evidence:**

No

**Claims Explanation:**

- The claim of establishing the “first generalizable CLU benchmark” appears too strong without sufficient reference support and is based only on very limited experiments.
- The submission provides promising empirical evidence, but it does not fully support several of the paper’s stronger claims. In particular, near-zero accuracy on forgotten classes and membership-inference accuracy near 0.5 do not show unlearning with precise, irreversible, or privacy-preserving settings.
- The proposed “knowledge leakage” measure compares performance across different class sets --- not tracking degradation on a fixed knowledge set.
- The code, preprocessing, class splits, and exact hyperparameters did not release or fully documented for reproducibility.

**Requested Changes:**

- The paper claims that each loss backpropagates based on its corresponding adapter, but all adapters contribute additively to the same network activations. Freezing the parameters of the other adapters prevents their parameters from changing. Please clarify and make changes accordingly.
- The model can achieve zero accuracy simply by systematically assigning forgotten samples to another class (at the same time, retaining considerable information about those samples). Please clarify and make changes accordingly.
- The MIA values near 0.5 and interprets them as evidence of privacy protection. Why? And need to clarify details about attack models (multiple attack initializations), architecture, and classes and sample sizes.
- Need to explain what “irreversible” means. What empirical test supports this claim?
- Not clear if the previously forgotten classes were tested again after later adaptation steps (to know forgetting persistence)/
- There are many formulation issues. Not consistent with the claims, including the sliding-window protocol, such as equations 1 and 3.
- The datasets are very limited. CASIA-Face100 contains only 100 selected identities. Need additional experiments on large and open datasets.

---

### Review · Reviewer_CDsE · 2026-07-18

**Summary Of Contributions:**

This paper addresses Continual Learning-Unlearning (CLU), where a pre-trained model must simultaneously retain relevant knowledge, acquire new knowledge, and remove designated knowledge over a sequence of adaptation requests. The authors identify “knowledge leakage,” i.e., the gradual degradation of retained capabilities over repeated learning-unlearning cycles, as a central challenge in this setting.

To address this issue, the paper proposes BID-LoRA, a parameter-efficient framework that freezes the backbone and introduces three dedicated LoRA pathways for retention, new knowledge acquisition, and forgetting. The forgetting pathway uses an escape-unlearning objective that moves representations of forget samples toward an embedding-space target selected to be distant from retained-class centroids, while the retention pathway uses replay and teacher-based embedding anchoring. The method updates roughly 5% of model parameters and is evaluated using a six-task sliding-window protocol on CIFAR-100 and CASIA-Face100. The reported results suggest that BID-LoRA can maintain competitive retain and new-class performance, achieve low forget accuracy, and remain relatively stable over multiple adaptation cycles.

Key strength:

1.The paper studies an important and practically motivated problem: jointly supporting continual learning and selective unlearning over repeated adaptation cycles.

2.The BID-LoRA design is conceptually simple and parameter-efficient. Separating retention, new learning, and forgetting into distinct adapter pathways is a reasonable way to reduce optimization interference.

3.The escape-unlearning objective is an interesting geometric mechanism for moving forget-class representations away from retained knowledge.

4.The six-task sliding-window evaluation goes beyond one-shot unlearning and provides a useful test of performance stability under repeated learn-forget operations.

5.The experiments cover both generic image classification and face recognition, and the ablations on LoRA rank, buffer ratio, pathway components, and escape scaling help motivate the proposed design.

Key weakness:

1.The privacy and unlearning evaluation is promising but not yet fully conclusive. Low forget accuracy, KL divergence, and membership-inference results provide useful evidence, but stronger attacks and more direct comparisons with retraining-based unlearning targets would strengthen the claim that forgotten information is no longer recoverable.

2.Some methodological details require clarification for reproducibility. In particular, the relationship between pathway-specific updates during training and the final merged model should be explained more precisely, including the sign used for the forget adapter in the merge operation and the treatment of classifier-head parameters.

3.The novelty should be positioned carefully relative to previous CLU work. The main contribution appears to be a parameter-efficient tri-pathway LoRA architecture with an escape-based forgetting objective, rather than the introduction of the CLU problem itself.

4.The experimental scope remains limited to two 100-class vision benchmarks, one fixed sliding-window protocol, and a 10% retain-data replay buffer. Additional datasets, task sequences, and larger-scale settings would strengthen the generalizability claims.

**Audience:**

Yes

**Audience Explanation:**

At least some members of the TMLR audience would likely be interested in this work. The paper addresses the intersection of continual learning, machine unlearning, and parameter-efficient adaptation, which is relevant to researchers studying lifelong learning, privacy-aware ML, foundation-model adaptation, and efficient fine-tuning. In particular, the paper highlights the practical challenge of handling repeated learning and deletion requests without full retraining, and proposes a simple LoRA-based design intended to reduce interference between retention, acquisition, and forgetting. The multi-cycle evaluation protocol and the face-recognition use case further make the findings relevant to researchers interested in realistic deployment settings where data and privacy requirements evolve over time.

**Broader Impact Concerns:**

The paper would benefit from a brief Broader Impact Statement. This is particularly important because face recognition is presented as a motivating application. While continual unlearning may support privacy and data-deletion requests, the same technology could also facilitate the continued deployment and expansion of surveillance or access-control systems. The paper should acknowledge this dual-use context.
In addition, the manuscript should clarify that its empirical unlearning metrics do not constitute a formal guarantee that sensitive information has been fully removed. Residual privacy risk may remain, especially under stronger extraction or inference attacks. Finally, because face-recognition performance can vary across demographic groups, the authors should discuss whether repeated learn-forget cycles could introduce or exacerbate subgroup disparities, and note the privacy and governance implications of maintaining a retain-data replay buffer.

**Claims And Evidence:**

Yes

**Claims Explanation:**

The main empirical claims are supported by reasonably clear and convincing evidence. BID-LoRA is evaluated across six sequential learning-unlearning tasks on both CIFAR-100 and CASIA-Face100, and the results consistently show low forget accuracy while maintaining competitive retained, new-class, and overall accuracy. Comparisons with several CLU baselines, together with ablations on LoRA rank, replay-buffer ratio, pathway components, and escape-point scaling, provide useful evidence that the proposed tri-pathway design and escape-unlearning objective contribute to the reported performance. The repeated-adaptation protocol also provides relevant evidence that the method can mitigate long-term knowledge leakage under the evaluated setting.

That said, the evidence is stronger for the paper’s benchmark-specific performance claims than for its broader privacy and generalizability claims. Although the paper includes an oracle retraining reference and a KL-based comparison, the unlearning evaluation would be strengthened by stronger and more diverse privacy attacks, as well as more comprehensive retraining-consistency analyses. Experiments on additional datasets, task configurations, buffer regimes, and multiple random seeds would also better establish robustness. These limitations do not negate the promising empirical results, but they should be acknowledged and addressed in a revision.

**Requested Changes:**

1.The paper should precisely explain how the three adapter pathways and their classifier-head parameters are handled during training and after merging. In particular, the apparent discrepancy between the additive adapter formulation and the negative sign for the forget adapter in Algorithm 1 should be resolved. This is important for reproducibility and for assessing whether pathway isolation persists in the final model.

2.The claims should clearly distinguish the demonstrated benchmark-level results from stronger claims of certified or broadly applicable privacy-preserving unlearning. The current experiments support promising empirical unlearning behavior, but do not by themselves establish a general privacy guarantee or broad real-world applicability.

Would strengthen the work

1.Add stronger and more diverse privacy or information-recovery attacks, and where feasible provide more comprehensive consistency comparisons against retraining-oracle models.

2.Include results over multiple random seeds with variance measures, along with wall-clock runtime, memory, replay-buffer storage, and total adaptation overhead. This would complement the reported tunable-parameter ratio.

3.Additional datasets, task orders, buffer sizes, and longer adaptation sequences would help establish the robustness of the method beyond the current two 100-class vision benchmarks and fixed sliding-window protocol.

---

### Review · Reviewer_ejRW · 2026-07-22

**Summary Of Contributions:**

This paper studies Continual Learning-Unlearning, where a model must simultaneously forget specified old classes, retain remaining old classes, and learn new classes across multiple adaptation cycles. The authors propose BID-LoRA, a parameter-efficient framework with three separate LoRA pathways for retention, new knowledge acquisition, and forgetting. The forgetting pathway uses an escape unlearning objective that pushes forget-class embeddings toward an escape point chosen to be far from retained-class centroids. Experiments on CIFAR-100 and CASIA-Face100 suggest that BID-LoRA achieves effective forgetting, stable retention, and good new-class learning while updating only about 5% of parameters.

**Additional Comments:**

Above

**Audience:**

Yes

**Audience Explanation:**

The paper addresses an emerging and practically relevant problem at the intersection of continual learning, machine unlearning, and parameter-efficient adaptation. TMLR readers interested in adaptive models, privacy-preserving learning, and efficient fine-tuning would likely find the CLU setting, the three-pathway LoRA design, and the multi-cycle evaluation protocol useful.

**Claims And Evidence:**

No

**Claims Explanation:**

1. The novelty and scope claims appear somewhat overstated. The paper claims to formalize CLU and establish a generalizable CLU benchmark, but the evaluation is limited to class-level learning/unlearning on two vision datasets. This is useful, but may not fully support claims about general CLU or broad continual adaptation.

2. The unlearning evidence is not fully convincing from a privacy perspective. Low forget accuracy, MIA near 0.5, and KL divergence are useful, but they do not necessarily prove that the influence of forgotten data has been removed. Stronger evaluations would help, such as retrain-from-scratch comparisons, probing recoverability of forgotten classes, and stronger membership inference analysis. The method relies on a retain buffer of at least 10% of the full retain data. This assumption weakens the privacy motivation, especially for face recognition.

3. There is a possible inconsistency in the method description. Equation 1 writes the combined adapter update with a positive forget adapter term, while Algorithm 1 merges the forget adapter with a negative sign.

**Requested Changes:**

1. Clarify the novelty and scope of the CLU formulation and benchmark. The paper should distinguish more carefully between prior CLU-like work, unified CL-UL approaches, and the claimed contribution of BID-LoRA.

2. Resolve the inconsistency in the forget adapter formulation. Equation 1 and Alg 2.

3. Strengthen the unlearning evaluation beyond forget accuracy, KL divergence, and MIA.

4. Discuss the privacy implications of requiring a retain buffer.

5. Temper broad claims such as “first generalizable CLU benchmark” unless supported by broader tasks beyond class-level adaptation on CIFAR-100 and CASIA-Face100.

6. The manuscript contains several typos and presentation issues that should be corrected. Examples include “shwon” → “shown,” “sqaured” → “squared,” “reply buffer” → “replay buffer,” “tablele” → “Table,” “recognistion” → “recognition,” and “utill” → “until.” The authors should carefully proofread the manuscript for grammar, spacing, and formatting issues.